# Chefs Saddle Up—Perceptions of Horse Meat as a Sustainable Gastronomic Alternative in France

**DOI:** 10.3390/foods14040638

**Published:** 2025-02-14

**Authors:** Céline Vial, Arnaud Lamy, Maxime Sebbane

**Affiliations:** 1Institut Français du Cheval et de l’Equitation, Pôle Développement Innovation et Recherche, 34000 Montpellier, France; 2MoISA, Université de Montpellier, CIHEAM-IAMM, CIRAD, INRAE, Institut Agro, IRD, 34000 Montpellier, France; arnaud.lamy@inrae.fr (A.L.); maxime.sebbane@supagro.fr (M.S.)

**Keywords:** meat consumption, horse meat, restaurant, chefs, market, sustainable food systems, gastronomy

## Abstract

This study investigates French chefs’ perceptions and knowledge of horse meat as a sustainable alternative in gastronomy. The research addresses the dual challenges of reducing the environmental impact of traditional meat production and reviving horse meat consumption, which has declined significantly in France due to cultural shifts and accessibility issues. Using semi-structured interviews with 12 chefs, including trainers and practicing professionals, the study explores their attitudes, personal consumption patterns, and professional willingness to incorporate horse meat into menus. The findings reveal that horse meat is valued for its nutritional and organoleptic properties, as well as its lower environmental impact compared to ruminant meats such as beef. However, barriers such as cultural taboos, limited knowledge, and insufficient culinary traditions impede its adoption. The chefs are categorized into three profiles—*connoisseur*, *pragmatic*, and *reluctant*—based on their personal and professional attitudes toward horse meat. The study concludes that promoting horse meat in select restaurants through education, recipe development, and targeted communication could enhance its acceptability and sustainability credentials. This work highlights the potential of horse meat to diversify protein sources in line with environmental and societal goals while addressing the specific needs of the equine sector.

## 1. Introduction

The restaurant sector plays a crucial role in the global food system, both as a major economic actor and as a driver of food consumption patterns. Eating out has become an integral part of modern lifestyles due to urbanization, changes in work routines, and evolving consumer preferences. In Europe, the foodservice industry represents approximately 12% of total food expenditures, with over 1.9 million restaurants [1]. The industry is also characterized by increasing out-of-home food consumption, with European households allocating an average of 30% of their food budgets to dining out, a trend that has been rising steadily over the past two decades [2]. Restaurants influence consumer food choices, shaping dietary habits through menu offerings, marketing strategies, and ingredient sourcing. However, this sector contributes substantially to the overall ecological degradation and faces significant environmental challenges that necessitate a shift towards more sustainable practices. These challenges include high levels of food waste, excessive meat consumption, and the overall carbon footprint associated with food production and preparation [3]. In this context, the shift toward sustainable gastronomy is gaining momentum, driven by consumer awareness and regulatory frameworks promoting responsible food practices, such as the European Green Deal’s Farm to Fork Strategy, that advocate for reducing meat consumption and incorporating alternative protein sources to mitigate environmental impacts [4].

Most European consumers are willing to adopt more sustainable dietary choices if adequately informed and provided with appealing options [5]. However, while household meat consumption is declining due to environmental and health concerns, consumers who reduce meat consumption at home tend to eat more meat when dining out [6]. This underscores the influence of chefs and restaurateurs in shaping dietary habits, highlighting the potential role of restaurants in promoting sustainable food alternatives.

The adoption of sustainable ingredients on restaurant menus, including non-traditional meats, aligns with these broader objectives. Among the underexplored alternatives to traditional red meat, horse meat presents an interesting case. Unlike beef and lamb, horses produce significantly less methane, require less feed and water, and provide leaner, high-protein meat with notable nutritional benefits [7]. Despite these benefits, cultural perceptions and limited availability have constrained its acceptance in contemporary European cuisine [8].

Thus, restaurants, as key players in shaping consumer food choices, could serve as entry points for reviving interest in horse meat while promoting sustainable dietary shifts. Despite the potential environmental and nutritional advantages of horse meat, little is known about how chefs perceive this product and whether they consider it a viable alternative to conventional red meat. This study explores French chefs’ perceptions, knowledge, and attitudes toward horse meat, addressing the following research questions:What are chefs’ personal spontaneous perceptions of the horse meat product?What are their professional experiences with this product?What do they know about the product and its production?Are there any differences among chefs ’attitudes towards horse meat?

By analyzing these aspects, this research contributes to the broader discourse on sustainable food systems and the role of chefs in influencing consumer choices, while exploring how alternative meats can be positioned within modern culinary practice.

The next part of this paper sets out the theoretical background. First, it contextualizes the issues by reviewing the impact of meat within food and restaurant systems. Second, attitudes towards horse meat are discussed from cultural and ethical perspectives in the French context. Third, the literature review explores how chefs can influence consumer perceptions in promoting sustainable alternatives to conventional red meat. The third part exposes the selected methodological framework, which is based on a qualitative approach. Results are presented in the fourth part and discussed in the fifth part, before the conclusion in the sixth part.

## 2. Literature Review

### 2.1. Sustainable Food and Restaurant Systems

The global agricultural and food system faces significant environmental and public health challenges. It currently accounts for a substantial share of greenhouse gas (GHG) emissions, contributing approximately 34% of global emissions [9]. Additionally, dietary quality is closely linked to the food system, with certain patterns associated with nutritional deficiencies and an increased prevalence of chronic diseases [10]. Addressing these dual challenges requires the promotion of sustainable diets, which emphasize reducing meat intake in favor of plant-based foods, such as fruits, vegetables, seeds, oilseeds, and legumes, especially in regions with high levels of meat consumption [11,12,13]. In addition to reducing overall meat consumption, it is essential to consider the type of meat consumed, as both species and farming practices significantly influence the environmental impact of meat production [14,15]. Consequently, recommendations for sustainable diets emphasize the importance of not only reducing meat consumption frequency, but also diversifying meat sources, including a shift toward alternative options such as camel, kangaroo, ostrich, and horse meat [16,17,18].

The restaurant system is linked to food systems and they share challenges with regard to the environment. The environmental impacts of the restaurant sector can be grouped into the following three main categories: direct impacts associated with energy and water consumption, emissions into air and water, and food resource waste; indirect impacts linked to the food supply chain, including food production, processing, and transport; and indirect impacts related to consumer behavior, such as food waste and the use of single-use items [19,20]. By analyzing the four main operational sub-activities of a restaurant—food supply, food storage, food preparation and cooking, and service activities—the research by Baldwin et al. (2011) [21] and Dai et al. (2020) [22] identifies food supply as the primary source of environmental impact. Specifically, because of its production processes, meat, and especially beef, stands out as the predominant contributor to the environmental impact of restaurants [22,23].

Finally, both food and restaurant systems need to take into account the environmental impact of meat. Among commonly consumed red meats, those from ruminant species—such as beef, lamb, and mutton—pose substantial environmental challenges due to their methane emissions and resource-intensive production [24,25,26]. In contrast, horse meat represents an exception among red meats, offering significant nutritional, environmental, and sensory benefits, as well as more favorable farming practices. Nevertheless, horse meat consumption is minimal, accounting for only 0.25% of the international meat market [7]. To explore the potential of horse meat as an alternative to conventional red meat, the cultural and ethical factors influencing food choices need to be explored.

### 2.2. French Cultural and Ethical Perspectives on Meat and Horse Meat Consumption

The French case is interesting for several reasons, firstly because meat consumption is high in this country; consequently, its reduction is an important issue. France is a Western nation characterized by high per capita meat consumption (83.3 kg/person/year; [27]). Nevertheless, there has been a slight decline in overall meat consumption over the past two decades (from 87.5 kg/person/year in 2003), even if these trends vary by species and type of meat product [27]. For example, poultry consumption has increased, while the intake of beef, pork, and lamb has declined. This high level of meat consumption is deeply rooted in French dietary and culinary traditions, where meat typically occupies a central place in meals and is often the primary component around which dishes are organized [28]. Culturally, meat consumption is imbued with symbolic meanings—such as vitality, masculinity, and strength—which have historically structured social relationships by differentiating gender roles and reinforcing communal bonds [29,30]. The primary motivation for meat consumption is the pleasure derived from it [31]. Among countries with a meat-centered food culture, France stands out for its particularly positive attitude toward beef [32], its relatively low willingness to reduce meat intake [33], and its more negative perception of meat-free diets, such as vegetarianism and veganism [32]. However, the importance of meat in dietary French habits varies by consumption context. For instance, meat consumption is notably higher in restaurants compared to home settings, with recent trends showing a decrease in home consumption and an increase in dining out, including at traditional restaurants, cafeterias, and fast-food outlets [34,35]. Similar patterns are observed in the UK and Germany, indicating that consumers who reduce their meat consumption at home often increase it in restaurants, highlighting the importance of the consumption context in shaping consumer meat preferences [6,36]. This trend may be influenced by the association of dining out with festive occasions, during which consumers may feel more inclined toward hedonic food choices [37].

Another interesting characteristic of France is that horse meat, which appears to be an interesting sustainable alternative to other red meats, is consumed in this country. First, on the environmental perspective, unlike ruminants, horses emit significantly less methane, approximately five times less than cattle [38,39,40]. Second, nutritionally, horse meat has a relatively moderate calorie density, lower than most other red meats [41]. It offers a high concentration of protein, comparable to that of beef and higher than that of some white meats [42], and it is particularly rich in essential amino acids. Compared to other red meats, horse meat is relatively low in fat, making it less fatty than beef and lamb. It has moderate cholesterol content and contains a higher proportion of unsaturated fats, including omega-3 and omega-6 fatty acids [7]. Horse meat is rich in important minerals such as iron, zinc, phosphorus, and potassium. It is particularly rich in heme iron [43], which is more easily absorbed by the body than plant-based iron. Horse meat is also a good source of several vitamins, mainly vitamins from the B group, such as B12 [44].

Third, horse meat production and consumption in France have a number of specific interesting characteristics. In terms of production, the breeding of draught horses for slaughter is conducted using extensive farming practices with small herds, addressing societal concerns about animal welfare. This approach also offers multifaceted environmental benefits, including land maintenance that complements that of ruminants and the preservation of sensitive or neglected areas. Additionally, it supports cultural heritage by preserving nine breeds of heavy horses and contributes to regional sustainability by fostering economic activity in rural areas [40,45,46]. Consequently, horse meat production represents an interesting alternative to other forms of red meat production from a sustainable perspective [42].

In terms of consumption, France has been eating horse meat since the 19th century, although its consumption is becoming less and less common. It has declined significantly, dropping by a factor of eight over the past 40 years, which is a more pronounced reduction than that of other meat types [8]. In 2021, approximately 7% of French households consumed horse meat, accounting for 0.1% of total meat purchases [47]. The horse meat consumption decrease is associated with ethical issues, including the emergence of a cultural taboo, as societal perceptions of horses have shifted from working animals to companions for leisure [8,17,42,48,49]. Horse meat consumption also varies according to cultural and situational contexts. Horse meat is still consumed regularly or occasionally in certain French regions, particularly among older generations and in family settings at home [8]. Furthermore, traditional supply channels, such as specialized butchers, are diminishing, with their numbers decreasing. While some supermarkets have begun offering horse meat, the product remains not actively promoted and can be difficult to find [50]. Moreover, a substantial part of the French population (around 15%) that does not currently consume horse meat is not morally opposed to it and is open to trying it, but does not do so because of limited opportunities due to the shrinking market and its lack of availability in restaurants [8,50]. These potential consumers view out-of-home settings, such as dining with friends and relatives or in restaurants, as opportunities to try horse meat. Among current consumers, the intention to consume is stronger at home or in restaurants. Consequently, restaurants present a promising opportunity for fostering the discovery and (re)consumption of horse meat while promoting it as a sustainable alternative on menus, especially given the influential role of chefs in shaping consumer choices.

### 2.3. The Role of Chefs in Shaping Sustainable Menus

Promoting “greener” menus has become a central objective for enhancing restaurant sustainability [3]. Several articles have explored strategies targeted towards consumers to encourage them in choosing less-meaty or vegetarian dishes [51,52,53,54]. Nevertheless, it is essential to understand the chefs’ perspective because they are responsible (in part or in full) for menus offered in the restaurants. Chefs can also play a crucial role in educating their staff and customers about the environmental impacts of food sourcing [55].

Firstly, chefs can focus on reducing overall meat consumption by incorporating more plant-based options into their menus. Studies have shown that increasing the availability of vegetarian and vegan dishes can significantly reduce meat consumption in restaurants [56,57]. It is particularly relevant to examine how they perceive and address sustainability issues related to meat [58,59,60]. Findings suggest that cultural context influences chefs’ approaches in the following ways: While chefs in Sweden [59] and the Netherlands [60] focus on reducing meat offerings, French chefs tend to prioritize measures such as sourcing quality ingredients (local, fresh, seasonal) and minimizing kitchen waste [58,61]. In France, chefs express both cultural and commercial hesitations regarding the reduction of meat on restaurant menus [58,61]. While developing strategies to promote plant-based options is essential, it is also interesting to explore complementary solutions, particularly the diversification of the types of meat dishes offered on the menu [3].

Secondly, chefs can emphasize the use of ethically sourced and sustainably produced meat. Research indicates that there is a growing consumer demand for meat products that are derived from small-scale, local farms that prioritize humane treatment of animals and sustainable farming practices [62]. By highlighting these practices in their menus and marketing, chefs can educate customers about the benefits of choosing ethically produced meat, which can also enhance the restaurant’s brand image and attract a conscientious clientele [62]. Leggett et al. (2021) [63] explore chefs’ perceptions of an alternative form of beef (Rocky Mountain Legume-Finished (RMLF) beef) that has superior environmental characteristics to traditional beef. In this case, taste and quality of the product take precedence over the environmental benefits and humane treatment of animals in order to meet the expectations of customers. These authors identify the following three main barriers to the adoption of this type of meat: limited availability; financial constraints, as this type of alternative can be more expensive, making it less profitable for the restaurant; and consumer preferences.

Finally, the restaurant industry faces several environmental challenges, one of them being the impact of red meat consumption. Chefs have a central role in this issue, as prescribers and through menu design. However, French chefs are not receptive to the idea of reducing meat and offering more plant-based alternatives. Consequently, it would be interesting to explore other ways of reducing the impact of conventional red meat. However, at this stage, nothing is known about chefs’ perceptions of alternative meats, despite their potential nutritional and environmental advantages over traditional meats. As horse meat is relevant in the French context, this study proposes an initial exploration of these perceptions.

## 3. Materials and Methods

A qualitative method, based on interviews with chefs, was used to collect data. A qualitative research approach is justified when the objective is to explore complex social phenomena, understand subjective experiences, or gain in-depth insights into behaviors, perceptions, and motivations [64]. Unlike quantitative methods, which seek to measure and generalize findings through numerical data, qualitative methods allow for a more nuanced and context-sensitive analysis of human interactions and decision-making processes. In the context of exploring chefs’ attitudes toward horse meat, a qualitative approach is thus particularly relevant, as it facilitates an understanding of their spontaneous perceptions, professional experiences, and knowledge about the product. Additionally, qualitative research can reveal underlying socio-cultural factors and ethical considerations that influence decision-making in professional kitchens.

### 3.1. Sample

The study is based on 12 semi-structured interviews conducted during the summer of 2020 in France with 7 chef trainers from the Institut Paul Bocuse (IPB), now known as Institut Lyfe, in the Rhône department, and 5 practicing chefs working in casual and fine-dining restaurants across the Rhône, Hautes-Alpes, Lot-et-Garonne, and Côte d’Or departments (Table 1). The interviews were conducted remotely via videoconference. The sample had an average age of 47 years (median: 52 years; range: 24 to 62 years) and included 2 women and 10 men (Table 1). All of the interviewed chefs began their training and careers at a very young age (between 13 and 16 years), with most expressing a lifelong passion for the field. The chef trainers had varying levels of experience at the IPB, ranging from less than 2 years to 17 years.

The chefs were recruited using a “snowball” approach, starting with the initial chef trainers interviewed and looking for the largest diversity of points of view among the chefs, which is encouraged by their different experiences and backgrounds. Recruiting practicing chefs was challenging because of their limited availability due to the demands of their profession. The interviews were conducted up to the saturation point (12 interviews). This number of interviews, even if low, is in line with the results of Hennink and Kaiser (2022) [65], who evaluated the sample size required for saturation in a systematic review, explaining that data saturation is generally reached between 9 and 17 interviews.

### 3.2. Interview Guide

The interview guide is structured around three main themes, each divided into sub-themes. After initial introductions by the interviewer, including an overview of the survey and interview conditions, the following first theme is introduced: the presentation of the interviewee. This section includes a brief history of the chef’s career and explores their vision of the profession (attraction to the profession, current views on the profession, sources of professional satisfaction and dissatisfaction), and their opinion on what constitutes a good menu and a successful restaurant dish. The second theme of the interview guide focuses on the chef’s perspective on meat, their mental representations of the product, personal meat consumption (including perceived advantages and disadvantages of consuming meat, and meats not consumed), and their professional relationship with meat (associations, aspects they appreciate or find lacking in meat preparations, and the skills and qualifications they believe are necessary). The third theme specifically addresses horse meat, examining the chef’s mental representations of the product and their perception of consumers/non-consumers, as well as the personal and professional knowledge and behavior of chefs.

By targeting two distinct groups of chefs—practitioners and trainers—the interview guide is tailored to the professional contexts of each group, incorporating specific questions about the role and teaching responsibilities of the trainers, as well as the characteristics of the offerings in their own establishments.

### 3.3. Collected Data and Their Processing

A consent form has been signed by every respondent. The collected data are not sensitive data and have been anonymized.

The interviews, which lasted on average 70 min (ranging from 35 to 100 min), were audio-recorded and then fully transcribed (140 pages, 94,000 words, 440,000 characters) (Table 1).

Data were analyzed using the **thematic analysis method**, as described by Braun and Clarke (2006) [66,67]. Thematic analysis is a widely used qualitative approach that allows for the identification, organization, and interpretation of patterns within qualitative data. This method was chosen for its flexibility in capturing both explicit (semantic) and implicit (latent) meanings in participants’ responses, making it particularly suitable for exploring chefs’ perceptions and experiences regarding horse meat.

The thematic analysis followed a structured phase process to ensure rigor and transparency, which is described as follows:-Familiarization with the data: the researchers repeatedly read through the transcribed interviews to immerse themselves in the data, taking initial notes and identifying potential patterns.-Generating initial codes: relevant excerpts were systematically coded using an inductive approach, ensuring that both expected and unexpected insights were captured.-Identifying, reviewing, and defining themes: The generated codes were examined and grouped into broader themes by identifying recurring patterns. These themes were refined through iterative discussions among the research team to ensure coherence and consistency.-Producing the analysis: the final themes were synthesized into the analysis, supported by illustrative verbatim excerpts from the interviews to enhance credibility and grounding in participants’ words.

The results are presented in a cross-sectional format, with thematic categories supported by verbatim examples to illustrate key findings.

## 4. Results

The results are split into four parts. The first three parts present transversal results concerning all of the interviewed chefs, which led to the fourth part, which underlines the variability among chefs through a typology.

### 4.1. A Specific Personal Relationship with the Horse

Discussions about horse meat invariably led the interviewed chefs to talk about the horse as an animal. The perceived image of the horse is strongly associated with positive values, emphasizing the current emotional bond between humans and horses (noble, affectionate, protective, man’s friend, sacred, faithful, vigorous, etc.). This image influences the respondents’ attitudes toward horse meat. There is a distinct opposition between appreciating the animal and consuming its meat, as obtaining and consuming horse meat implicitly implies the death of the animal and a betrayal of the values associated with it.

“I think that in France, the horse is sacred. It’s a pet or a racing animal, which is put on a pedestal by French people. Consequently, this animal is something that we won’t kill and that we won’t eat, that’s the vision I have of it”. (Chef 9)

The relationship between horses and humans is described by the chefs as a longstanding one throughout history, characterized by notions of friendship, companionship, and loyalty. According to the respondents, this relationship takes different forms, involving the horse in the universe of work, including agricultural, military, therapeutic, transport, riding, racing, sport, and emotional support.

“In Bourgogne, there is a reintroduction of draft horses to work in the vineyards. Instead of machines, it’s really making a comeback with biodynamics, it brings new life by reintroducing species that were already there”.(Chef 10)

“To return to the topic of horses, we can describe them as companions, helping and caring for some people”.(Chef 12)

This particular relationship with horses contrasts the one reserved for cows, which often serves as a reference when comparing animals, meats, or the link between an animal and its meat. Among the various stances taken by the chefs regarding horses and their meat, the following horse/cow opposition stands out as particularly interesting:

For Chef 4, a horse meat enthusiast, the relationship between humans and horses throughout the horse’s life adds value to the meat product, which does not apply in the case of cows, as shown in the following quotation:

“The way that horses serve humans even after death is, to me, truly magnificent, because the horse gives everything until the very end. It lived for us, and it died for us. There’s no such thing with the cow”.(Chef 4)

Chef 6 questions this exclusive application of a special status to horses, as follows:

“At home with my wife and two daughters, it’s out of the question to imagine putting horse meat on their plates. It’s not possible, we respect horses too much. Traditionally even though we’re farmers—well my daughter is—but she won’t eat horse. There are so many other things to eat, that’s her philosophy. She’s the one who says it, there are so many other things to eat, so why are we obliged to kill a horse? But you can apply that logic to all animals: beef—why do we kill it?”(Chef 6)

Horse meat is frequently compared to other animals regarding the acceptability of its consumption (rabbit, elephant, snake, ostrich, kangaroo, pig). It is also compared to animals whose dietary status is evolving (rabbit) or deemed non-consumable (dog, cat, guinea pig). Rabbits, for instance, are considered both a pet and an edible animal, whereas dogs, cats, and guinea pigs are considered non-consumable pets, as in the following quote:

“I think horses have become pets, and very few people today eat their dogs and cats”.(Chef 8)

In these comparisons, the perception of the animal differs based on the nature of the relationship between humans and the species, as well as the cognitive abilities attributed to the animal. Horses, like cats and dogs, are often perceived by their owners as species possessing cognitive abilities similar to those of humans. This perception may lead to a greater tendency to anthropomorphize horses.

### 4.2. A Product Rarely Offered in Restaurants

Some chefs note the current presence of horse meat in the catering industry, including in restaurants they frequent or have visited in the past. While certain types of restaurants—such as meat-focused establishments, themed restaurants, specialized fast-food outlets, and palaces—are perceived as more likely to include horse meat on their menus, they are often not the restaurants where the respondents currently work. For some respondents, the integration of horse meat into menus depends on a revival of culinary trends and greater awareness about this meat.

Beyond these anecdotal experiences, many chefs emphasize the absence of horse meat in restaurants. They say that it can even be considered as having no place on restaurant menus. The predominant opinion appears to be that horse meat should not be offered in restaurants. On a personal level, the majority of respondents agree that they will never put horse meat on their restaurant menus.

“I’ve never seen it on a menu! In all my years of experience, and I’ve moved around a lot, I’ve never come across it”.(Chef 2)

“Yes, it’s a personal philosophical choice. For me, it’s not a meat that should be offered in restaurants”.(Chef 6)

According to the perceptions of the chefs interviewed, the special emotional bond between horses and humans contributes to a problem of acceptability and social norms that exert pressure against consuming horse meat. As a result, few consumers show interest in this product, and its consumption is no longer considered part of French gastronomic culture, as demonstrated by the following quotes:

“I don’t think so, I think it’s purely cultural that we don’t eat horse here. Or maybe we used to, but not anymore”.(Chef 8)

“I think that, for example, unlike in other cultures, eating horse meat is not part of French culture”.(Chef 9)

A number of chefs spoke of horse meat consumption as rooted in the past, being a bygone experience no longer perpetuated in the present.

Chefs characterize the typical horse meat consumer as an elderly meat-lover, living in a rural area or specific region, curious, and a “bon vivant”, a person who enjoys the pleasures of life, especially good food and drink. This type of consumer remains a minority compared to the general population.

“So, it’s people who like meat, a bit older, and often bon vivants and epicureans”.(Chef 3)

On the contrary, the profile of non-consumers of horse meat is more varied, including uninformed individuals, discerning eaters, young people, vegetarians, and equestrian practitioners. Above all, respondents emphasize that non-consumers make up the majority of the French population. Some chefs also point out that this non-consumption aligns with broader food trends, such as localism, meat reduction, and vegetarianism.

“I don’t think horse meat will come back into fashion. I believe it’s a thing of the past. I think that today, there is a general shift away from meat consumption. Poultry isn’t as affected, but red meat consumption is declining. I think people are becoming aware that they don’t have to eat red meat every day. On top of that, there’s the issue cost, and the controversy surrounding slaughterhouses and livestock farms has encouraged people to become vegetarians, flexitarians, or to reduce their meat consumption”.(Chef 11)

### 4.3. A Lack of Knowledge Among Chefs, but an Openness to the Product

The interviewed chefs displayed the following characteristics, suggesting an openness and potential interest in horse meat: they highly value meat in general; they are open to innovation and curious about new products; and they have an interest in “healthy” menus and show concern for animal welfare, the environment, and local produce.

Regarding the product itself, chefs discuss the price. Horse meat is considered expensive, more so than beef.

“No, I don’t eat it anymore, because it’s hard to find, and when you do find it in supermarkets, it’s very expensive. Horse meat has become a luxury; for the same weight of meat…, if you compare a 150 g filet of horse to the same weight of beef, it’s not the same price, it’s very expensive”.(Chef 2)

Conversely, chefs seldom consider the ecological attributes of horse meat due to their limited knowledge of the industry and production methods.

Chefs describe the advantages of this meat based on the various intrinsic attributes they associate with it.

With regard to organoleptic properties, verbatim comments mainly focus on the meat’s taste, color, and texture. Its smell is not mentioned.

The descriptions of horse meat’s taste emphasize its “good” flavor, strong and distinct for some, compared to beef (stronger) or game.

Color is regarded as an important distinguishing feature of horse meat, noted for its red hue (described as black by some respondents). Consequently, lighter-colored horse meat, such as that produced by French young draught horses, may be less appealing to chefs who favor horse meat, as demonstrated by the following quotation:

“Yes, for me, color is appealing. It differentiates meats, and above all, confirms that the meat is horse—it’s very bloody. Because that’s what color represents, it’s not anything else”.(Chef 3)

The texture is compared to that of beef, with its tenderness receiving particular emphasis.

The nutritional qualities of horse meat are widely acknowledged and associated with health benefits. It is regarded as energizing, nourishing, low in fat, and rich in protein, vitamins, and minerals, including iron.

“Hyper-nourishing meat—energetic, bloody and good for health—it was considered a pharmaceutical food, a type of food that was viewed as healing by past generations. We used to give horse meat to pregnant women and to children when they were weak. So, this meat is very bloody, very powerful, and provides significant benefits to the human body”.(Chef 3)

However, these properties are rooted in a bygone era, resembling a “grandmother’s remedy” more than a modern food-medicine.

For some respondents, horse meat can also evoke concerns about health risks, either due to the origin of the slaughtered horses or the 2013 lasagna crisis in France, where horse meat was found in products that were supposed to be exclusively beef-based. This incident, described as consumer fraud and deceit, led to a lack of trust and increased focus on meat traceability.

“For me, horse meat is not appealing, mainly because of all the trafficking that has occurred”.(Chef 7)

“I think there’s an enormous amount of work to do on this subject to prove that horses are healthy and not injected with hormones… Today, we know about all the treatments given to animals, but in the past we didn’t”.(Chef 7)

However, this knowledge often takes on the status of belief, raising questions about chefs’ actual knowledge.

“In my opinion, it can be very good, because horses aren’t fat. […] It’s just a prejudice I have, because to me, a horse is protein, it’s dynamic. So, I subconsciously think that horse meat will give me energy, or at the very least, it won’t be bad for me”.(Chef 1)

The chefs acknowledge their lack of knowledge about the horse meat sector, which remains largely invisible within the agri-food industry, including the types of horses slaughtered, slaughter conditions, origin of consumed animals, existence of a draught horse breeding sector for butchery, potential supply sources, and the reality of current consumption.

“I have no idea what breed of horse, what type they are, or how they’re raised or slaughtered”.(Chef 10)

Although respondents rarely mention production methods, they are more likely to question sourcing methods, particularly purchasing locations.

Horse butcheries are the most commonly cited locations, but references to them are more focused on the past than the present, reflecting their gradual disappearance in modern times.

“In France, I remember there were a few horse butcheries when I was a kid. Now I think they are very rare in France. I don’t even know if there’s one in Lyon—I don’t think so”.(Chef 12)

Chefs also mention other locations, such as supermarkets, traditional butchers, halal butchers, and tripe butchers.

Sourcing is also a concern for restaurants, with worries about the potential difficulty of finding suitable suppliers. Purchasing criteria are also mentioned, such as meat provenance and the importance of favoring local products.

Some chefs assume that horse meat mainly comes from reformed animals, older horses that are sent to slaughter, such as in the following quote:

“I don’t think there are many farms producing this meat, I think it’s mostly slightly older animals that are sent to the slaughterhouse, that’s my impression”.(Chef 5)

Chefs are also curious about the French draught horse breeding industry and its meat products, which they are unfamiliar with. They assume that this meat is of high quality.

“You’ve just mentioned draught horses with red meat, and now I’m curious to learn more about this type of meat”.(Chef 1)

“I think our product is certainly of higher quality than the products we import”.(Chef 11)

With regard to the culinary use of horse meat, various preparations are mentioned, particularly steak (grilled, sautéed, rare) and tartare (raw, minced, Caesar-style), which are two traditional ways of consuming this meat, although they overshadow the “roasted” form. At the same time, some chefs emphasize the importance of serving this meat rare. Other possibilities mentioned include bourguignon, ham, stew, parmentier, gardiane, and hamburgers.

Nevertheless, some respondents note the absence of horse meat recipes, such as in the following quote:

“We don’t have any traditional French recipes for horse meat that are recognized, created by great chefs, or considered part of our gastronomic identity”.(Chef 9)

This last point reflects the training of chefs, as horse meat is absent from student apprenticeship programs. According to some trainers, horse meat is a product that could potentially shock students.

“I think what would shock students more is working with horse meat. If horse is written on the label, I’m doomed. That would go viral on Twitter, Facebook and Insta, and then we’re finished. The Institut Paul Bocuse is committing horse genocide, there’s 80 kg of it… [Laughs]”.(Chef 1)

Finally, the results are summarized in Table 2, which presents the obstacles and levers to the use of horse meat in restaurants, according to the 12 interviewed chefs.

### 4.4. A Variability Among the Surveyed Chefs

First and foremost, the chefs differ in their individual consumption practices regarding horse meat. Consumption experiences vary, with some respondents describing having eaten horse meat in restaurants, abroad, or in specific social settings. The variety of living contexts and social ties influences horse meat (non-)consumption practices, particularly those rooted in the family home during childhood.

“I’m not aware of any horse butcheries nearby. It’s true that I don’t eat horse meat, but I did when I was a kid. I’m 57 now, but I remember when I was 12–14, my mother used to cook it at home. Not every week, but maybe 3–4 times a year we ate horse”.(Chef 11)

“I’ve never been brought up to eat horse meat”.(Chef 7)

Thus, this personal relationship with horse meat appears to be linked to the level of knowledge about the product and to the professional openness to it. As a result, three profiles seem to emerge among the chefs interviewed (Table 3).

#### 4.4.1. *Connoisseur* (2 Chefs/12: Chefs 3 and 4)

These chefs’ personal consumption of meat in general is assumed. They are personally horse meat enthusiasts, with a positive view of the product and knowledge about it. They are likely to offer it or already feature it on their menu. While they perceive some limitations to the use of this meat in restaurants, they are less restrictive than other types (e.g., offering it as a daily special or as a suggestion, rather than integrating it directly into the menu)

“Yes, as a regular dish of the day, I put it on my menu as a suggestion or as a special”.(Chef 3)

These more experienced and older chefs have a traditional approach to cooking, where meat is a key component of culinary identity.

#### 4.4.2. *Pragmatic* (8/12 Chefs: Chefs 1, 2, 6, 7, 9, 10, 11, and 12)

These chefs are rare consumers, former consumers, or non-consumers of horse meat; however, they are curious to learn more about the product on a personal level.

“Why not try it in a restaurant, even if I’m disappointed and don’t finish my plate? If they manage to sell it well, I’ll buy it”.(Chef 1)

“We were asked to taste it. I’ve also had the opportunity to eat it a few times. It’s been several years since I’ve eaten any”.(Chef 12)

Their knowledge of the product varies and is incomplete. They perceive the following major limitations to its use in restaurants: moral limits (cultural acceptability among customers), economic limits (expensive product), and technical limits (sourcing methods, preservation/maturation, and expertise), which collectively dissuade them from offering this meat on their menus.

“I’ve never put it on my restaurant menu. If that’s a question you’re asking, no I’ve never offered horse”.(Chef 6)

The following two sub-groups stand out:Young, inexperienced chefs, curious and potentially attracted to this meat, but with little or no knowledge of the product.Experienced chefs, who take a broader approach when considering integrating this product into their menus.

#### 4.4.3. *Reluctant* (2 Chefs/12: Chefs 5 and 8)

Non-consumers or former consumers, these chefs are personally reluctant to consume horse meat. This reticence contrasts with their relationship with other esteemed meats, such as beef, where the separation between the animal and the meat is less distinct. For example, in the case of Chef 5, the more or less personal connection and the images associated with the animal appear to influence their perception, viewing the horse as inedible—both the animal and its meat—while beef is seen as edible, both in its live and meat forms, as follows:

“When I see a horse, I don’t feel like eating it, I feel more like riding it. But when I see an Aubrac cow, I think about eating it”.(Chef 5)

Knowledge of the product is limited and often tied to past consumption. These chefs stress their emotional bonds with horses, which they view as pets, and oppose the presence of horse meat on restaurant menus.

This typology shows that chefs are strongly influenced by their personal tastes, food histories, and consumption practices. Exposure to horse meat, particularly during childhood, and continued consumption in adulthood, seem to shape the chef’s knowledge and consideration of the product in their profession. *Connoisseurs* are more likely to offer the product in restaurants compared to *Pragmatic* and especially *Reluctant* chefs. Finally, a personal connection with the animal appears to influence a similar professional stance, blurring the distinction between personal identity and professional identity as a chef.

## 5. Discussion

### 5.1. Overall Contributions

This research provides insights by bridging the domains of sustainable gastronomy and cultural perceptions of alternative meats. The results contribute to the ongoing discourse on how chefs can act as mediators between sustainable food systems and consumer behavior. Previous studies have emphasized the pivotal role of chefs in promoting sustainability through menu design, ingredient sourcing, and consumer education [3]. While previous works have focused on plant-based transitions [57], this study offers a complementary perspective by exploring the potential of diversifying red meat sources through less conventional options, like horse meat, which addresses both environmental and societal goals. By categorizing chefs into distinct profiles—*Connoisseur*, *Pragmatic*, and *Reluctant*—the study offers a nuanced framework for understanding how personal practices, knowledge, and cultural norms shape professional openness to including horse meat on menus. Thus, our findings align with and expand upon the existing research on chefs’ attitudes toward sustainable practices in the foodservice sector.

The categorization of chefs in this study (*Connoisseur*, *Pragmatic*, and *Reluctant*) mirrors findings by Lamy and Costa (2023) [58], who identified varying levels of acceptance and resistance among chefs toward incorporating sustainable practices, often shaped by personal and cultural perceptions. The limited knowledge and visibility of horse meat, as highlighted in this study, align with findings from Popoola et al. (2022) [18], who noticed that unfamiliarity with alternative meats hinders their adoption. This research also corroborates the prior literature by emphasizing the substantial cultural and emotional barriers, where chefs’ personal relationships with horses and the extent to which equids are anthropomorphized plays a critical role in their acceptance as food [17]. In contrast, other alternative meats, such as kangaroo or bison, are more readily accepted, with their consumption primarily driven by considerations of taste rather than cultural or ethical concerns [49].

This research thereby enriches the broader literature on sustainable gastronomy by calling for targeted education, recipe development, and communication strategies tailored to chefs’ varying levels of openness and expertise.

### 5.2. Consumption Determinants: Toward a Possible Development?

The horse meat market is unique in that the non-consumer population, which is significantly larger, dominates the consumer population. This dominance negatively affects the market’s legitimacy, both at home and in restaurants, presenting a challenge that must be addressed. The following two key determinants are particularly difficult to influence: the socio-cultural relationship with horses and eating habits; these factors are central to a system of norms and values shared by a large segment of the French population. As a result, the potential for growth in horse meat consumption largely depends on targeting individuals for whom horse meat is still considered acceptable for human consumption and who can integrate it into their eating habits (red meat consumers) [8]. The same pattern is observed among chefs, whose professional attitudes are influenced by their personal consumption habits.

The characteristics of the offer can be redefined along the following four axes, for which different levers can be activated:Product-related characteristics:
The French sector, which raises draft horses for meat in outdoor conditions, meets the needs of tradition, localism, ecology, and animal welfare, by supporting the preservation of endangered breeds, limiting the import of meat, and offering meat that emits fewer greenhouse gases (methane) than ruminant meats like beef.The organoleptic characteristics offer significant hedonic potential in terms of taste and texture, which have yet to be fully exploited at the culinary level.The nutritional properties of horse meat (source of essential amino acids, minerals, and vitamins) offer substantial potential to address current health challenges, as highlighted by several authors [41,44].As highlighted for chefs, previous studies [49] reported a lack of consumer knowledge on the subject. Communication about horse meat is currently limited, leaving the field open to its detractors. Consequently, the first lever of action would be to develop communication campaigns to inform consumers about the benefits of this meat, particularly in terms of its environmental impact, breeding conditions, and organoleptic and nutritional properties. Although it may be disadvantaged by current nutritional recommendations aimed at limiting the consumption of animal proteins, horse meat can stand out due to the quality of its nutritional profile and its potential as a sustainable alternative in the diversification of animal protein sources.
The high price of horse meat can be an obstacle to its consumption, especially since it is primarily consumed by the working classes. However, restaurant consumption could be adapted to overcome this constraint, as people are generally more willing to pay a high price to eat in restaurants than at home.Distribution is also a challenge, given the low visibility of the offer in supermarkets, the gradual disappearance of horse meat butchers, and the extremely limited availability in restaurants. However, with better visibility of the offer for both at-home and out-of-home consumption, along with the rise of alternative channels, horse meat shows potential for development.

### 5.3. A Communication Strategy for Catering Professionals

As we have seen, offering horse meat in restaurants represents a consumption opportunity, not only by current consumers, but also for potential consumers who are interested in discovering the product. Promoting the presence of this product in commercial catering therefore seems like a relevant strategy to develop the market.

For stakeholders of the equine sector, a business-to-business strategy targeting catering professionals seems necessary. This strategy would involve disseminating knowledge about the sector and the product, highlighting the potential interest of a segment of the French population, and emphasizing the advantages of the product. Cooking techniques should also be promoted, as Popoola et al. (2022) [18] attributed the unpopularity of horse meat to respondents’ lack of awareness of horse meat-based recipes. Restaurants that focus on meat products (from independent steak houses to famous chains) should be prioritized. Furthermore, using horse meat could help chefs address the contemporary challenge of creating more sustainable cuisine without reducing meat consumption, as Lamy et al. (2023) [58] found that chefs are unaware of this potential.

However, it is important to communicate in a differentiated manner, as shown for consumers [50], among whom certain groups are a priority depending on their general attitude toward meat and horse meat in particular. Indeed, some chefs (the *Reluctant* ones) impose a moral taboo on consuming this meat, so it is advisable to avoid targeting them with communication that they would likely disregard, and which could even provoke opposition.

*Pragmatic* chefs should be a target for communication, aiming to first develop their knowledge of the product (regarding the local draught horse sector in particular), and second, reassuring them about the potential commercial viability of the product for customers. Given that the chefs’ positioning on this product is closely linked to their personal relationship and consumption of it, offering horse meat discovery and tasting sessions would be a valuable approach.

Finally, although *Connoisseur* chefs are a minority, they are convinced and ready to promote horse meat in restaurants. Consequently, it would be interesting to further enhance their knowledge of the product, develop a recipe booklet, facilitate access to the product, and encourage them to offer it more as a daily special or a suggestion, rather than incorporating it directly into their menus. These efforts could be part of coordinated actions by the horse meat sector, such as a promotional campaign or creativity sessions, like the one organized with the IPB, which resulted in the creation of a recipe booklet.

## 6. Conclusions

There are obstacles and levers to the consumption of horse meat in France today. First, improved visibility, promotion, and availability of the product could increase its purchase frequency among consumers. Additionally, potential consumers who are currently non-consumers represent 15% of the French population [8,50], suggesting significant growth potential for this market. These individuals are more likely to try horse meat when dining out than at home, influencing the demand development strategy. In this perspective, chefs, who are influencing prescribers, are interesting to study. The results of this research show that some chefs are interested in the product and might offer it on their menus. However, they lack information about the product, its distribution channels, and a rich culinary repertoire for horse meat. This lack of knowledge is evident not only among current chefs, but also in the training programs for future chefs, leading to an inability to effectively promote horse meat.

Although the level of knowledge remains low, the intrinsic characteristics of horse meat are generally perceived favorably by chefs, who highlight its organoleptic qualities (taste, color, and texture), as well as health and nutritional benefits (an excellent source of protein because of an interesting amino acid profile, but also of minerals (especially iron and zinc), and vitamins, while being relatively low in fat but with a qualitative fatty acid profile).

This research underlines that personal convictions influence chefs’ professional attitudes, which is a particularly interesting element to consider in other future studies. The biggest barrier remains the question of the cultural acceptability of horse meat, which operates on the following two levels with chefs: directly through their personal views, and indirectly through perceptions of customer opinions. *Reluctant* chefs are distinguished by their personal non-acceptance of the product, while *Pragmatic* chefs emphasize the cultural non-acceptability of the meat. In both cases, these chefs contribute to the taboo surrounding horse meat in the out-of-home catering sector by limiting its presence in their establishments.

On the other hand, *Connoisseur* chefs are aware of the cultural reluctance of some customers, but position themselves as defenders of the practice and are more optimistic about its inclusion in restaurants. This position aligns with the role some chefs take in advocating for fresh, local, and traditional products, or those from the extensive meat sectors to which horse meat can belong.

In a managerial perspective, this study sheds light on useful recommendations for socio-professional and institutional actors of the sector in order to better understand the opportunities and limits of the market, as well as for training and practicing chefs who would like to act in favor of the promotion of horse meat in the restaurant industry. Indeed, in a highly competitive catering market, offering horse meat can serve as a distinctive element for certain types of French restaurants, sparking the curiosity of current and potential consumers while addressing societal expectations for commercial catering in terms of diversifying animal protein sources and supporting more sustainable gastronomy.

This research presents some limitations linked with a small number of respondents, the limited representativeness of the sample that does not cover all French regions, and including few women. Moreover, the results only rely on chefs’ perceptions and beliefs, as we did not offer them direct access to horse meat. Consequently, the paper paves the way towards future research on possible alternatives to red meat in a sustainable perspective through chefs’ influence. It would be interesting to deepen this study with a quantitative survey relying on a large sample representative of the chef population, to open the study to other kinds of meat, to observe the consequence of giving access to the product itself, or to information and knowledge about the product.

Finally, the results obtained have to be taken with a pinch of salt due to methodological limitations. Nevertheless, they offer interesting first insights into managerial and theoretical advances for gastronomical sustainability.

## Figures and Tables

**Table 1 foods-14-00638-t001:** Characteristics of the sample.

Characteristic	Detail
Total number of interviews	12
Number of practicing chefs	5
Number of chef trainers	7
Respondent age	Average of 47 years (median: 52 years; range: 24 to 62 years)
Respondent gender	2 women and 10 men
Duration of interviews	Average of 70 min (range: 35 to 100 min)
Length of transcriptions	140 pages, 94,000 words, 440,000 characters

**Table 2 foods-14-00638-t002:** Obstacles and levers to the use of horse meat in restaurants (according to the 12 interviewed chefs).

	Obstacles	Levers
Cultural	According to chefs, horse meat is not appealing to many consumers and suffers from cultural acceptability issues	
	Some chefs consider horse meat a product that was consumed in the past but is no longer part of French gastronomic culture	
Personal	Chefs have a personal relationship with horses that leads to a greater anthropomorphization of horses	
Professional	Chefs point to the lack of horse meat recipes	Meat holds an important place in general for chefs
	There is no instruction on horse meat for chefs-in-training	Chefs are open to innovation and new products, and are interested in healthy menus, animal welfare, the environment, and local produce
	Chefs’ lack knowledge about horse meat, including its production, distribution, and consumption	Chefs are curious about the French breeding sector of draft horses for butchery
	Chefs have limited awareness of the environmental benefits of horse meat compared to other red meats like beef	
Structural	The price of horse meat is high	Chefs are aware of the intrinsic advantages of horse meat: nutritional properties, color, taste, and texture
	There is a lack of availability and accessibility of the product	According to chefs, horse meat could be adapted to certain types of restaurants: meat-focused establishments; themed restaurants; specialized fast-food outlets; and palaces

**Table 3 foods-14-00638-t003:** The three chef profiles.

	*Connoisseur* (N = 2)	*Pragmatic* (N = 8)	*Reluctant* (N = 2)
Personal practices regarding horse meat	Regular or occasional consumers Positive view of the product	Rare consumers or non-consumers Curiosity towards the product	Non-consumers or former consumers Emotional connection to the horse which is viewed as a pet Personally reluctant to consume horse meat
Knowledge of horse meat	Knowledge of the product	Varied and incomplete knowledge of the product	Limited knowledge of the product
Professional openness to horse meat	Perceive some non-prohibitive limits to the use of this meat Willing to offer it or already offering it on their menu	Perceive significant limitations to the use of horse meat: moral, cultural, economic, and technical (sourcing, preservation/maturation, and expertise)	Reluctant to offer horse meat in restaurants

## Data Availability

The data presented in this study are available on request from the corresponding author (the data are not publicly available due to privacy or ethical restrictions).

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
