# Peer review of "Chefs Saddle Up—Perceptions of Horse Meat as a Sustainable Gastronomic Alternative in France"

_foods, 2025, doi:10.3390/foods14040638_

Round 1
Reviewer 1 Report
Comments and Suggestions for Authors
After reading the manuscript, I have the following concern toward this study.
I wonder what is the cost to raise horses as daily meat and can the reduced emission from producing horse meat replace the cost?
This study lacks a literature review section, which reduces the theoretical foundation of this study.
Regarding the theree chef profiles, I sense the result is quite common sense and the interview results did not enrich the current understanding.
The overall sample of the interview is not large and I suggest using questionnaire to collect more data to confirm the suitability of the profiling.
While analyze the interview data, the study needs to describe what detailed method it has used and what are the steps involved.
Regarding the discussion, it is more report style instead of academic style and the discussion focuses on giving some practical considerations while ignoring the theoretical one.
From my perspective, cultural perception of meat largely determines customers' choice of meat, which cannot be easily changed. The aforementioned high cost may be another barrier to promote horse meat in our daily consumption. Thus, the study needs more justifaction and literature review in order to show the meaning to conduct this study.
Author Response
Reviewer 1:
Dear reviewer 1,
Thanks for your kind comments and suggestions. Our answers are included in bold and italic in the following text. In the revised manuscript, changes are reported in red. We did our best to answer your comments and suggestions and we hope to have met these requests.
After reading the manuscript, I have the following concern toward this study.
I wonder what is the cost to raise horses as daily meat and can the reduced emission from producing horse meat replace the cost?
It is difficult to estimate the economic cost to produce horse meat per kilo due to a lack of data because these are niche productions and there is a large variety of cases. Consequently, it is difficult to compare horse meat production to other animal productions. On the other hand, there is scientific data showing that horses emit five times less methane than cattle (as explained in our paper). Our approach here is exploratory and aims to pave the way for avenues of consumption of this type of meats as more sustainable alternatives, it therefore opens the way to other avenues of research, some of which may actually have this objective in order to assess the viability of this production.
This study lacks a literature review section, which reduces the theoretical foundation of this study.
Thank you for this comment. We reviewed the introduction and the theoretical framework, giving it more structure (creating an introduction and a theoretical framework with 3 sub-sections) and we expanded the literature on several points (for example the role of chefs in proposing more sustainable menus). The research objectives are now more salient in relation to the literature.
Regarding the three chef profiles, I sense the result is quite common sense and the interview results did not enrich the current understanding.
Indeed, the results are not surprising in their entirety, but they are interesting in several purposes.
First, few studies focus on chefs ‘perceptions of meat-related issues or sustainability, so these results add to this emerging literature. We added elements to explain that more clearly in the introduction part.
On the other hand, in the results, it is interesting to see that some chefs can be quite reluctant to this meat use in restaurants whereas other are not. Moreover, the fact that personal convictions seem to influence chefs’ professional attitudes is a particularly interesting element to consider in other future studies. We added elements to explain that more clearly in the conclusion part.
The overall sample of the interview is not large and I suggest using questionnaire to collect more data to confirm the suitability of the profiling.
Thank you for this comment. We added a paragraph concerning the limits and perspectives of the study in the conclusion, which details these points among others.
While analyze the interview data, the study needs to describe what detailed method it has used and what are the steps involved.
Thank you for this remark. We added a paragraph describing the qualitative data analysis. Data was examined using a thematic analysis based on the method proposed by Braun and Clarke (2006):
“The data collected was analyzed using the thematic analysis method (Braun and Clarke, 2006; Nowell et al., 2017). The thematic analysis was carried out in several stages (familiarization with the data, creation of an initial coding plan, grouping and organizing the themes, analyzing the verbatims within the same theme). The themes were identified at semantic and latent levels”.
Regarding the discussion, it is more report style instead of academic style and the discussion focuses on giving some practical considerations while ignoring the theoretical one.
Thank you for your valuable feedback regarding the discussion section. To address your concern about the lack of theoretical focus, we added a new subsection entitled "4.1 Overall Contribution" within the discussion. This section emphasizes the theoretical implications of the study, situating our findings within broader academic frameworks and highlighting their relevance to existing literature.
By doing so, we aim to balance the practical considerations already discussed with a more rigorous theoretical perspective, ensuring that the discussion aligns with an academic style while contributing meaningfully to the scholarly discourse.
From my perspective, cultural perception of meat largely determines customers' choice of meat, which cannot be easily changed. The aforementioned high cost may be another barrier to promote horse meat in our daily consumption. Thus, the study needs more justification and literature review in order to show the meaning to conduct this study.
Thank you for this comment. We reviewed the theoretical framework in order to better justify the objective of this research. Based on this framework, we identified the challenge of reducing and diversifying red meat in diets and menus for environmental reasons. In the French context, research has shown that chefs are not in favour of reducing meat on their menus through a more plant-based cuisine. It is therefore interesting to identify their perception of an alternative type of red meat. Still in the French context, horse meat is an interesting example of alternative red meat because it has environmental advantages and it is a meat that is nowadays rarely consumed without being culturally excluded from consumption. This context therefore justifies exploratory research to identify whether chefs, with aspirations for sustainability and the preservation of meat on their menus, could use horse meat as an alternative to traditional red meats.

Reviewer 2 Report
Comments and Suggestions for Authors
“Chefs saddle up - Perceptions of horse meat as a sustainable gastronomic alternative” is a well-written and interesting paper. However, I have some comments and concerns that I would like to see addressed. Please see below.
The study involves interviews to 12 chefs in France recruited using a snowball procedure. This is not a randomised procedure, which limits the full perspective of what should be the view of the chiefs. Are these 12 chiefs representative of the Universe you wanted to study? Also, you need to specify the universe of study. Is this France? A particular region or regions in France? It would be important to define the universe and include a word describing it in the title. A limitations section at the end of the discussion also highlighting future avenues for research would help address the issue.
Lines118-119, 331-333, 494-505, 554-557. While referring to the nutritional quality and benefits of horse meat you refer to protein and fat. Probably you could include in the introduction a table comparing the nutritional composition of the major types of meat for comparison (pork, beef, chicken, horse,… eventually mouton/lamb). Please be aware that the protein content of these types of meat will be very similar therefore saying that horse meat is rich in protein is not a valid argument. Horse meat has however an amino acid protein profile that is richer in some essential amino acids. This should be your argument worth mentioning. In terms of fat, horse meat has indeed lower quantities when compared to beef, however, it is worth mentioning the composition in terms of fatty acids as horse fat has higher quantities of unsaturated fatty acids therefore with health benefits. In the conclusion instead of referring to proteins (plural) you should refer to protein (singular) and then add a differentiation in terms of amino acid profile. The same for the fatty acid profile of fat.
Minor issues:
Line 125: not sure if animal welfare is the right societal concern here. Probably conservation of genetic diversity?
In methods: I can see that informed consent was obtained. However, you also need to state the ethical approval of the study. Please include a reference to this in your methods section.
Author Response
Reviewer 2:
Dear reviewer 2,
Thanks for your kind comments and suggestions. Our answers are included in bold and italic in the following text. In the revised manuscript, changes are reported in red. We did our best to answer your comments and suggestions and we hope to have met these requests.
“Chefs saddle up - Perceptions of horse meat as a sustainable gastronomic alternative” is a well-written and interesting paper. However, I have some comments and concerns that I would like to see addressed. Please see below.
The study involves interviews to 12 chefs in France recruited using a snowball procedure. This is not a randomised procedure, which limits the full perspective of what should be the view of the chiefs. Are these 12 chiefs representative of the Universe you wanted to study?
Thank you for this comment. In line with the qualitative methodology, we did not seek to make the population of studied chefs representative, but we rather aim to have a diversity of points of view among the chefs, which could be encouraged by their different experiences and backgrounds. We recruited chefs until we reached data saturation, i.e. until the last interviews revealed no new themes in relation to the first ones identified, in line with the results of Hennink and Kaiser (2022), who evaluated the sample size required for saturation in a systematic review, explaining that data saturation is generally reached between 9 and 17 interviews. We added explanation on these points in the methodological part of the paper.
Also, you need to specify the universe of study. Is this France? A particular region or regions in France? It would be important to define the universe and include a word describing it in the title.
The study focuses on French chefs and the respondents come from several regions of France. This is explained in the text and we added, as you kindly suggested, the term “in France” in the title of the paper.
A limitations section at the end of the discussion also highlighting future avenues for research would help address the issue.
Thank you for this comment. We added a paragraph concerning the limits and perspectives of the study at the end of the conclusion, which details this issue among others.
Lines118-119, 331-333, 494-505, 554-557. While referring to the nutritional quality and benefits of horse meat you refer to protein and fat. Probably you could include in the introduction a table comparing the nutritional composition of the major types of meat for comparison (pork, beef, chicken, horse,… eventually mouton/lamb). Please be aware that the protein content of these types of meat will be very similar therefore saying that horse meat is rich in protein is not a valid argument. Horse meat has however an amino acid protein profile that is richer in some essential amino acids. This should be your argument worth mentioning. In terms of fat, horse meat has indeed lower quantities when compared to beef, however, it is worth mentioning the composition in terms of fatty acids as horse fat has higher quantities of unsaturated fatty acids therefore with health benefits. In the conclusion instead of referring to proteins (plural) you should refer to protein (singular) and then add a differentiation in terms of amino acid profile. The same for the fatty acid profile of fat.
Thank you for your valuable feedback regarding nutritional properties of horse meat. As you suggested, we insisted on the interesting amino acid protein profile of horse meat instead of it protein content which is similar to other red meats. In the same perspective, we added elements regarding the fatty acid profile of horse meat. We also added elements concerning its calorie density, mineral and vitamin content (lines 121-130). However, we decided not to include a comparative table of different meats, as it is not the core subject of our paper.
In the discussion (lines 578-579) and the conclusion (lines 650-653) we modified the text as you proposed, to refer to protein (singular) and to the amino acid profile, instead of proteins.
Minor issues:
Line 125: not sure if animal welfare is the right societal concern here. Probably conservation of genetic diversity?
Thank you for this remark. We moved this part of the sentence about societal concerns on animal welfare, which is linked to the fact that the breeding of draught horses for slaughter is conducted using extensive farming practices.
In methods: I can see that informed consent was obtained. However, you also need to state the ethical approval of the study. Please include a reference to this in your methods section.
This subject was discussed directly with the Editor of the Journal, who had the same concerns. We explained that this research doesn't need an Ethics Committee or Institutional Review Board approval because we collected anonymous information and this information is not sensitive data. Nevertheless, a Data Management Plan was realized and a consent form has been signed by every respondent. This consent form (in French and translated in English) has been sent to the Editor of the Journal.
Finally, we added a sentence in the methodological part of the paper to explain that.

Reviewer 3 Report
Comments and Suggestions for Authors
Dear authors
I find your work innovative, and well conducted and I am congratulating for it.
However, in order for the manuscript to be improved and accepted, from my side of view, major revisions in the text can be done, requiring minnor extra effort, as follows:
1. Simple summary: should be canceled (there is no need for it!)
2. Introduction: lines 83-96can be cancelled dicreasing the size of the section
3. End of indroduction: the exact study scope should be presented in bullets (reffering to the 4 levels of grouped questions [3.1 - 3.4] presented below
4. Materials and Methods: Whis is the "quantitative method" you used for your research BASED ON EXISTING LITERATURE?!!!!
5. The used BASIC questinnaire should be presented at a supplementary file
6. 2.1: when was the study taken place?
7. The results of 3.1, 3.2, 3.3. and Table 2 should be rewritten in a way similar to the 3.4 results and Table 3 (which is not only covering ALL 12 answers by the reviews, but also even categorizing them IN ANN EXCELLENT WAY!)
8. The discussion should be rewritted discussing actual results such as the chiefs categories (reluctant / pragmatic / connoisseur) in comparison with related existing literature
9. The conclusion then should be rewritten also including: limitations of the study, future suggested studies, groups of public and private entities to whom the study will be usefull etc.
Author Response
Reviewer 3:
Dear reviewer 3,
Thanks for your kind comments and suggestions. Our answers are included in bold and italic in the following text. In the revised manuscript, changes are reported in red. We did our best to answer your comments and suggestions and we hope to have met these requests.
I find your work innovative, and well conducted and I am congratulating for it.
However, in order for the manuscript to be improved and accepted, from my side of view, major revisions in the text can be done, requiring minnor extra effort, as follows:
- Simple summary: should be canceled (there is no need for it!)
The simple summary has been suppressed.
- Introduction: lines 83-96 can be cancelled decreasing the size of the section
We changed the first part of the manuscript, reducing the size of the introduction and adding a part on the literature review with three sub-sections, which gives more structure to the paper. The paragraph you suggested to remove has been shortened.
- End of introduction: the exact study scope should be presented in bullets (referring to the 4 levels of grouped questions [3.1 - 3.4] presented below
Thank you for this valuable comment that helped us give more structure to the paper. At the end of the introduction, we detailed the sub-objectives of the research in the form of bullet points, as you suggested. Moreover, we added a paragraph presenting the structure of the paper.
- Materials and Methods: Whis is the "quantitative method" you used for your research BASED ON EXISTING LITERATURE?!!!!
Thank you for this comment. We added in the methodological part more explanations about the methodology and references on the method used to analyse our qualitative data (thematic analysis, Braun and Clarke 2006, Nowell et al., 2017).
- The used BASIC questionnaire should be presented at a supplementary file
Thank you for your suggestion regarding the inclusion of the questionnaire. To address this, we added the questionnaire as a supplementary file. This includes both the original French version of the interview guides (one for practitioners and one for trainers) and their English translations, ensuring accessibility and transparency for a broader audience. We hope this addition provides greater clarity and utility for understanding the methodology used in the study.
- 2.1: when was the study taken place?
Thank you for this comment. The study was carried out during summer 2020, and we have indicated the date in the methodology.
- The results of 3.1, 3.2, 3.3. and Table 2 should be rewritten in a way similar to the 3.4 results and Table 3 (which is not only covering ALL 12 answers by the reviews, but also even categorizing them IN ANN EXCELLENT WAY!)
Thank you for your suggestion. We appreciated the fact that you value the part 3.4 of the results. We were not able to modify results of 3.1, 3.2, 3.3. and Table 2 to present them in the same way as the part 3.4, because the three first parts of the results give rise to the typology that is presented in part 3.4. Nevertheless, we added a sentence at the beginning of the result part explaining that (“The result part is split in four parts. The three first parts present transversal results concerning all the interviewed chefs, which led to the fourth part that underlines the variability among chefs through a typology.”) and we added in the text and in the title of Table 2 that the data comes from the 12 interviewed chefs.
- The discussion should be rewritted discussing actual results such as the chiefs categories (reluctant / pragmatic / connoisseur) in comparison with related existing literature
Thank you for your valuable feedback regarding the discussion section. To address your concern about the lack of theoretical focus, we added a new subsection entitled "4.1 Overall Contribution" within the discussion. This section emphasizes the theoretical implications of the study, situating our findings within broader academic frameworks and highlighting their relevance to existing literature.
By doing so, we aim to balance the practical considerations already discussed with a more rigorous theoretical perspective, ensuring that the discussion aligns with an academic style while contributing meaningfully to the scholarly discourse.
- The conclusion then should be rewritten also including: limitations of the study, future suggested studies, groups of public and private entities to whom the study will be usefull etc.
Thank you for these suggestions that enabled us to improve the conclusion part. As you proposed, we detailed the targets of the managerial recommendations that come from this research. We also added a paragraph concerning the limits and perspectives of the study.

Round 2
Reviewer 1 Report
Comments and Suggestions for Authors
After reading the revised manuscript, I find the study has been improved based on reviewers' comments, but I still have the following concerns toward the revised manuscript:
The introduction is too simple and more information should be given about gastronomic alternatives, horse meat and even people's food choice.
In literature review, some paragraphs are too short and should be rearranged to form some more complete paragraphs including topic sentences and supporting sentences. A short summary and critique of past studies should be done at the end of this section.
The steps involved in thematic analysis should be explained in a more detailed way instead of just naming the steps as there are also several different ways to do thematic analysis and the study needs to demonstrate why the way they chose is the most proper one.
The study explained they used the qualitative method, but did not explain why qualitative one is better than the quantitative one, which is a very crucial point in method choice.
Regarding the other comments I raised in the first round, authors have put them in the limitation section and I hope these limitations can be really addressed in future studies.
Author Response
Dear reviewer 1,
We are happy that you considered the paper improved in this second round of revision. Thanks for your new kind comments. Our answers are included in bold and italic in the following text. In the revised manuscript, changes are reported in red. We did our best to answer your suggestions and we hope to have met these requests.
Best regards.
The authors.
After reading the revised manuscript, I find the study has been improved based on reviewers' comments, but I still have the following concerns toward the revised manuscript:
The introduction is too simple and more information should be given about gastronomic alternatives, horse meat and even people's food choice.
Thank you for this comment. We have substantially expanded this section by integrating data that underscore the restaurant sector's significance within the European food system. Additionally, we have included insights into the environmental impact of restaurant operations and explored the potential of horse meat as a sustainable alternative. Consumer habits and preferences are also discussed.
In literature review, some paragraphs are too short and should be rearranged to form some more complete paragraphs including topic sentences and supporting sentences. A short summary and critique of past studies should be done at the end of this section.
Thank you for this remark. We rearranged this part of the paper to group paragraphs together and not let too short paragraphs alone. A paragraph has been added at the end of the literature review to summarize this part and underline the lacks coming from past studies.
The steps involved in thematic analysis should be explained in a more detailed way instead of just naming the steps as there are also several different ways to do thematic analysis and the study needs to demonstrate why the way they chose is the most proper one.
Thank you for this comment. In the methodological part, this revision provides a clearer justification for the choice of thematic analysis and details each step more thoroughly.
The study explained they used the qualitative method, but did not explain why qualitative one is better than the quantitative one, which is a very crucial point in method choice.
To address your concern, we revised our section 3 to explicitly contrast the strengths of qualitative research with the limitations of quantitative methods in this context.
Regarding the other comments I raised in the first round, authors have put them in the limitation section and I hope these limitations can be really addressed in future studies.

Reviewer 2 Report
Comments and Suggestions for Authors
Dear authors,
Thanks for addressing the issues raised. I am pleased to recommend the publication of this manuscript.
Author Response
Dear reviewer 2,
Thanks for your help in improving our paper. We are happy that you consider the paper now suitable for publication.
Best regards.
The authors.
Reviewer 3 Report
Comments and Suggestions for Authors
The revized manuscript can be published as it is now
Author Response
Dear reviewer 3,
Thanks for your help in improving our paper. We are happy that you consider the paper now suitable for publication.
Best regards.
The authors.